# Humanized L184Q Mutated Surfactant Protein C Gene Alters Alveolar Type 2 Epithelial Cell Fate

**DOI:** 10.3390/ijms25168723

**Published:** 2024-08-09

**Authors:** Krishan G. Jain, Yang Liu, Runzhen Zhao, Preeti J. Muire, Jiwang Zhang, Qun Sophia Zang, Hong-Long Ji

**Affiliations:** 1Department of Surgery, Stritch School of Medicine, Loyola University Chicago, Maywood, IL 60153, USA; kjain2@luc.edu (K.G.J.); rzhao6@luc.edu (R.Z.); qzang@luc.edu (Q.S.Z.); 2Burn and Shock Trauma Research Institute, Stritch School of Medicine, Loyola University Chicago, Maywood, IL 60153, USA; pmuire@luc.edu; 3Department of Orthopedics and Rehabilitation, Stritch School of Medicine, Loyola University Chicago, Maywood, IL 60153, USA; 4Infectious Diseases and Immunology Research Institute, Stritch School of Medicine, Loyola University Chicago, Maywood, IL 60153, USA; 5Department of Cancer Biology, Oncology Institute, Cardinal Bernardin Cancer Center, Loyola University Medical Center, Maywood, IL 60153, USA; jzhang@luc.edu; 6Departments of Pathology and Radiation Oncology, Loyola University Medical Center, Maywood, IL 60153, USA

**Keywords:** alveolar type 2 epithelial cell, L184Q mutation, differentiation, proliferation, re-alveolarization, surfactant protein C

## Abstract

Alveolar type 2 epithelial (AT2) cells synthesize surfactant protein C (SPC) and repair an injured alveolar epithelium. A mutated surfactant protein C gene (*Sftpc^L184Q^,* Gene ID: 6440) in newborns has been associated with respiratory distress syndrome and pulmonary fibrosis. However, the underlying mechanisms causing *Sftpc* gene mutations to regulate AT2 lineage remain unclear. We utilized three-dimensional (3D) feeder-free AT2 organoids in vitro to simulate the alveolar epithelium and compared AT2 lineage characteristics between WT (C57BL/6) and *Sftpc*^L184Q^ mutant mice using colony formation assays, immunofluorescence, flow cytometry, qRT-PCR, and Western blot assays. The AT2 numbers were reduced significantly in *Sftpc*^L184Q^ mice. Organoid numbers and colony-forming efficiency were significantly attenuated in the 3D cultures of primary *Sftpc*^L184Q^ AT2 cells compared to those of WT mice. Podoplanin (PDPN, Alveolar type 1 cell (AT1) marker) expression and transient cell count was significantly increased in *Sftpc*^L184Q^ organoids compared to in the WT mice. The expression levels of CD74, heat shock protein 90 (HSP90), and ribosomal protein S3A1 (RPS3A1) were not significantly different between WT and *Sftpc*^L184Q^ AT2 cells. This study demonstrated that humanized *Sftpc*^L184Q^ mutation regulates AT2 lineage intrinsically. This regulation is independent of CD74, HSP90, and RPS3A1 pathways.

## 1. Introduction

Surfactant protein C (SPC), uniquely produced by alveolar type 2 (AT2) cells, is a critical molecule for respiratory health. It reduces alveolar surface tension, prevents alveolar collapse, and facilitates gas exchange [1]. The lung injury caused by bacterial or viral infections like influenza and severe acute respiratory syndrome coronavirus 2 (SARS-CoV-2) kills AT2 and AT1 cells and impairs gaseous exchanges [2]. AT2 cell death correlates with reduced SPC levels [3]. AT2 cells are distal lung progenitor cells that proliferate and transdifferentiate into mature AT1 cells to regenerate the epithelial barrier and restore gaseous exchange [4,5]. Recent studies have reported that AT2 passes through a transition state to differentiate into mature AT1 cells. These transitional AT2 cells expressed elevated cell cycle arrest markers, downregulated AT2 markers, and modestly upregulated AT1 markers. These findings suggest that the accumulation of dysfunctional transitional AT2 cells with suppressed AT1 differentiation may be the specific regenerative defect underlying the pathogenesis of human IPF [6].

Mutations in the *Sftpc* gene, such as *Sftpc*^L188Q^, *Sftpc*^I73T^, and *Sftpc*^C121G^, have been associated with neonatal respiratory distress syndrome (NRDS) and idiopathic pulmonary fibrosis (IPF) [7,8]. To date, more than 70 SPC mutations have been identified in IPF patients or children with childhood interstitial lung disease (chILD) [9]. In families carrying an *SFTPC* mutation, incidental cases of lung cancer have been reported [10]. *SFTPC*^L188Q^ (mutation analog to L184Q in mice), a mutation in the BRICHOS domain of precursor SPC protein (proSPC), inhibits its trafficking to the Golgi body for posttranslational modification. Mistrafficking induced degradation of mutant proSPC, resulting in a deficiency of mature functional SPC [11]. Expressions of *Sftpc*^L184Q^ mutation (equivalent to *SFTPC*^L188Q^ in humans) were associated with decreased lung regeneration in adult mice and disrupted the growth of AT2 cells in the lungs of neonatal mice, leading to a permanent reduction in the number of AT2 cells, which are the source of SPC in the lungs [11]. In our recent study, we found that surfactant proteins were reduced in exosomes derived from AT2 cells in the bronchoalveolar lavage (BAL) fluid of acute respiratory disease syndrome (ARDS) patients [12].

CD74 is expressed on the surface of AT2 cells and alveolar macrophages [13]. It has been approved as a biomarker for lung infections and diseases [14,15]. The expression of CD74 seems to be a double-edged sword in the lungs. On the one hand, CD74 levels increase in response to lung injuries and infections like coronavirus disease 2019 (COVID-19), suggesting a potential role in defense mechanisms [16,17]. On the other hand, some viruses like influenza can decrease CD74 in AT2 cells [18]. This complexity is further highlighted by the observation that AT2 cells in adenocarcinoma expressed high levels of CD74 [19].

HSP90 is a key player in inflammation, and HSP90 inhibition suppresses inflammation [20]. High HSP90 levels are linked to inflammatory lung diseases like systemic sclerosis, pulmonary fibrosis, and lung cancer [21,22,23,24]. HSP90 inhibition protected hamsters from SARS-CoV-2-induced lung injury [25]. HSP90 inhibitors 17-DMAG, 17-AAG, and AUY-922 effectively protected the lung from inflammation, restored pulmonary endothelial barrier function, and reduced lung fibrosis in mice [21,22,26,27].

RPS3A is a component of the eukaryotic 40S small ribosome subunit that serves as a chaperone and regulates protein translation [28,29]. High levels of RPS3A are linked to transformed cells and tumors [30,31]. Studies have shown that RPS3a can influence cell growth, differentiation, death (apoptosis), and response to drugs [32]. RPS3A was associated with the risk of coronary artery diseases in humans [33]. A recent study demonstrated that RPS3A is involved in lipopolysaccharide (LPS)-induced pro-inflammatory cytokine production in RAW264.7 cells, but there is no evidence for this connection in lung repair [29]. Therefore, the roles of CD74, HSP90, and RPS3A1 in lung health appear multifaceted, requiring further investigation to understand their impacts fully. All three of these pathways are known to be involved in inflammation and injury, as well as regulating cell proliferation and differentiation [20,29,34].

We hypothesize that *Sftpc*^L184Q^ reduces AT2 to AT1 transdifferentiation by halting AT2 cells in a transitional state and directing AT2 lineage during alveolar repair. Accumulation of transitional cells impairs lung repair and induces fibrosis. To model the deficiency of SPC in ARDS lungs, we utilized humanized mice carrying the *Sftpc*^L184Q^ mutation. To study the mechanism by which SPC deficiency alters the regenerative potential of AT2 cells, we compared the characteristics of WT and *Sftpc*^L184Q^ AT2 lineage in the lung organoid model.

## 2. Results

### 2.1. Sftpc ^L184Q^ Mutation Altered AT2 Yield and Marker Protein Expression In Vivo

To evaluate the effects of *Sftpc*^L184Q^ on the AT2 cell population in vivo, we compared epithelial cell adhesion molecule (EPCAM^+^, Gene ID: 4072) AT2 cells between WT (C57BL/6) and *Sftpc*^L184Q^ mice (Figure 1A). Our findings revealed a significantly (*p* < 0.05) reduced number of EPCAM^+^ AT2 cells in *Sftpc*^L184Q^ lungs compared to in WT control lungs (Figure 1B). The purity of AT2 cells was confirmed by immunofluorescence staining (Appendix A). Western blot data showed a significant (*p* < 0.05) reduction in AT2 cell marker, proSPC expression, and elimination in mature SPC in *Sftpc*^L184Q^ lungs compared to WT control lungs (Figure 1C,F). In contrast, the expression level of another AT2 cell marker, surfactant protein B (SPB) was not altered (Figure 1D,G). The PDPN protein, a marker protein for AT1 cells, was significantly (*p* < 0.05) reduced in *Sftpc*^L184Q^ lungs compared to WT controls (Figure 1E,H). Full-sized blots are available as Appendix A.

### 2.2. Sftpc^L184Q^ Mutation Downregulated AT2 Lineage In Vitro

Given the SPC reduction in the harvested AT2 cells from *Sftpc*^L184Q^ mice, we reasoned that *Sftpc*^L184Q^ mutation regulates AT2 cell stemness. We isolated EPCAM^+^ AT2 cells from WT and *Sftpc*^L184Q^ mice and grew them as 3D organoids in matrigel (Figure 2A). Comparative analysis of brightfield images (Figure 2B–D) showed that *Sftpc*^L184Q^ AT2 had a significant reduction in proliferative organoids (*p* < 0.0001) and colony-forming efficiency (*p* < 0.0001) compared to WT cultures. As colony formation is characteristic of actively dividing cells, we evaluated EdU incorporation into the DNA of actively dividing AT2 cells in proliferative organoids. *Sftpc*^L184Q^ organoids contained a significantly lower number of EdU^+^ cells compared to WT controls (*p* < 0.0001, Figure 2E,F).

### 2.3. Sftpc^L184Q^ Disrupted Organoid Size and Structure

To identify the effects of reduced AT2 proliferation and transdifferentiation into AT1 on *Sftpc*^L184Q^ organoids’ sizes and structures, we performed confocal imaging of immunolabeled organoids. As observed in confocal images (Figure 3A) of proliferating organoids, *Sftpc*^L184Q^ organoids were smaller in size relative to WT organoids. The smaller *Sftpc*^L184Q^ organoids also possessed fewer SPB^+^ AT2 cells compared to larger WT organoids. Confocal images of differentiated organoids (Figure 3B) revealed a change in the structures of *Sftpc*^L184Q^ organoids. In comparison to WT organoids, *Sftpc*^L184Q^ organoids displayed a small lumen covered by a thick surface layer of transitional AT2 cells. In contrast, WT organoids displayed a larger lumen surrounded by a single layer of AT2 and differentiated AT1 cells. Western blot (Figure 3C,D) analysis demonstrated a significant increase in PDPN protein expression in *Sftpc*^L184Q^ organoids compared to WT organoids, possibly due to the accumulation of SPB^+^ and PDPN^+^ transitional AT2 cells. Interestingly, SPB protein levels were not significantly different between WT and *Sftpc*^L184Q^ organoids. Full-sized blots are available as Appendix A. 

### 2.4. Sftpc^L184Q^ Halted Maturation of Transitional AT2 Cells into AT1 Cells

AT1 differentiation is a hallmark of alveolar regeneration. Given the effect of *Sftpc*^L184Q^ mutation on AT2 proliferation, we wanted to further confirm the implication of *Sftpc*^L184Q^ mutation on AT2 transdifferentiation into AT1 in the 3D organoid model; therefore, we checked the effect of mutated proSPC and deficiency of mature SPC on AT2 proliferation and AT1 differentiation in organoids. We dissociated differentiated organoid colonies into single cells and labeled them with AT2 and AT1 markers, i.e., SPB and RAGE-specific antibodies, and subjected them to flow cytometry to analyze AT2 lineages (Figure 4A). Analysis of flow cytometry results demonstrated that most of the cells in both WT and Sftpc^L184Q^ groups differentiated into AT1 and transitional cells. Only a few AT2 cells were observed. *Sftpc*^L184Q^ organoids had significantly reduced RAGE^+^ AT1 (*p* < 0.001) cell counts compared to WT organoids (Figure 4B–D). Intriguingly, we observed a significant increase in the SPB^+^ RAGE^+^ transitional AT2 cell (*p* < 0.001) count in *Sftpc*^L184Q^ organoids compared to WT organoids (Figure 4D).

### 2.5. Sftpc^L184Q^ Did Not Alter CD74, HSP90, or RPS3A1 Expression in Organoids

The mechanisms of AT2 proliferation and differentiation are closely related to lung inflammation and injury. Given the associations of CD74, HSP90, and RPS3A1 expression and function in inflammation and injury, we assessed the expression levels of these three proteins in AT2 cells. Intriguingly, the *Sftpc*^L184Q^ mutation demonstrated no significant differences in the mRNA and protein levels of CD74, HSP90, and RPS3A1 compared to the WT control. CD74 and RPS3A1 protein levels were reduced in *Sftpc*^L184Q^ organoids compared to WT organoids, but the differences were not significant (Figure 5A–H). Interestingly, we observed reduced expression levels of CD74 and RPS3A1 mRNA in *Sftpc*^L184Q^ organoids compared to those in the WT controls, but the differences were insignificant. Full-sized blots are available as Appendix A.

## 3. Discussion

This study suggests that 3D organoids derived from AT2 cells harboring the *Sftpc*^L184Q^ mutation can serve as a human disease model to reveal how this mutation can lead to human AT2 dysfunction and provide a platform for testing potential treatments for the disease. Previous studies found that constitutive expression of *Sftpc*L^184Q^ disrupts the growth of AT2 cells in the lungs of neonatal mice. This leads to a permanent reduction in the number of AT2 cells, which are important for lung repair in adult life [11]. Our findings from adult mice demonstrated that *Sftpc*^L184Q^-mutated lungs possess a significantly reduced number of AT2 cells compared to WT control lungs. Studies in mice with *Sftpc* ^Δexon4^.transgene or the *Sftpc*^C121G^ allele showed death of AT2 cells, abnormal lung development, and death shortly after birth [35]. However, in humans, mutated SPC did not involve postnatal death [11]. Our adult *Sftpc*^L184Q^ mouse model exhibits suppressed mutated proSPC protein and lacks processed SPC, recapitulating the misprocessing occurring in vivo in humans [11].

The mutation of the *Sftpc* gene can have significant consequences for lung health [36]. This study examined how the *Sftpc*^L184Q^ mutation influences AT2 cell lineage. Organoids originating from *Sftpc*^L184Q^ AT2 cells displayed significant reductions in colony numbers and colony-formation efficiency. *Sftpc*^L184Q^ mutation disrupts AT2 proliferation and transdifferentiation processes, leading to fewer mature AT1 cells and accumulation of transient AT2 cells compared to in WT control organoids. Furthermore, our in vitro EdU incorporation assay data reinforce this concept, and our observations are consistent with previous research indicating decreases in AT2 cell expansion and AT1 differentiation in postnatal *Sftpc*^L184Q^ lungs and organoid models [11]. They reported that treatment with antioxidants restored AT1 differentiation in *Sftpc*^L184Q^-mutated organoids, to levels comparable to WT controls [11]. This suggests that aberrant *Sftpc* gene expression may cause increased oxidative stress within AT2 cells [11]. These mature SPC-deficient cells might represent an intermediate stage in the alveolar epithelium’s physiological or pathological renewal process.

To understand the underlying mechanism regulating SPC-mediated AT2 proliferation and differentiation, we compared the expression levels of CD74, HSP90, and RPS3A1 proteins between WT and *Sftpc*^L184Q^ organoids. CD74, HSP90, and RPS3A1 serve as protein-folding chaperones and are known to be involved in inflammation, injury, and regulating cell proliferation and differentiation [20,29,34]. Influenza infection reduced CD74 and proSPC expression in mice lungs [18]. Hydroxychloroquine (a drug used for ILD therapy) increased the expression level of HSP90 by 81% in *Sftpc*^I73T^ mutant MLE-12 cells [37]. IPF patients and a mouse model of pulmonary fibrosis overexpressed RPS3A1 [38]. Therefore, we investigated the expression levels of CD74, HSP90, and RPS3A1 in organoids to reveal their significance in AT2 cell proliferation and differentiation. Intriguingly, our RT-PCR and immunoblotting data revealed no significant differences in the expression levels of CD74, HSP90, and RPS3A1 between *Sftpc*^L184Q^ and WT AT2 organoids. Our study suggests that additional cellular chaperones may play roles in protein folding within lung cells. These chaperones would help in ensuring that proSPC folds correctly and undergoes maturation to produce functional SPC. Currently, little is known about how chaperones like HSP70, calreticulin, and calnexin function in this process. Furthermore, the potential involvement of the immune system in re-alveolarization cannot be excluded. Stressed AT2 cells might trigger an immune response that contributes to lung fibrosis associated with SPC deficiency. This study sheds light on the role of the SPC in AT2 lineage regulation in *Sftpc*^L184Q^ mutant organoids. However, immunofluorescent tracking in vivo is necessary to assess *Sftpc*^L184Q^ AT2 lineage directly. The precise mechanism by which *Sftpc*^L184Q^ regulates alveolar regeneration remains an intriguing area for further investigation. This understanding of mechanisms could identify potential therapeutic targets for the treatment of ARDS and other lung diseases.

## 4. Materials and Methods

### 4.1. Animals

The wild-type (WT) C57BL/6j mice were obtained from Jackson Laboratory (The Jackson Laboratory, Bar Harbor, ME USA). The *Sftpc*l^184Q^ mutant mice, bred on a C57BL/6j background, were sourced from Dr. Timothy E. Weaver and Dr. Jeffrey A. Whitsett’s lab in the Department of Pediatrics at Cincinnati Children’s Hospital Medical Center. All mice were housed in a pathogen-free facility with a 12 h light/dark cycle, and they had ad libitum access to food and water. Paired WT and *Sftpc*^L184Q^ mutant mice, age-matched (2–4 months), both male and female, were used for experiments. These procedures were conducted following approval from the Institutional Animal Care and Use Committees of the University of Texas Health Science Center in Tyler and Loyola University Health Sciences Division in Chicago.

### 4.2. Fluorescence-Activated Cell Sorting (FACS) Mediated AT2 Cell Isolation

AT2 cells from mice were isolated as published previously [39]. Briefly, mice were anesthetized with Ketamine/Xylazine (100/8.5 mg/kg), and the mice were exsanguinated. Subsequently, the lungs were perfused with Dulbecco’s phosphate-buffered saline (DPBS, Gibco, Grand Island, NY, USA #14190250) to eliminate blood. Then, lungs were infused with 1.5 mL of 50 U/mL dispase (Corning Inc., Corning, NY, USA#354235) solution through the trachea, followed by a low-melting-point agarose to prevent dispase solution leakage. The lungs were then further incubated in 3 mL of 50 U/mL dispase solution for 45 min at 25 °C. After enzymatic digestion, cells were released from the lung by delicately teasing it in advanced DMEM/F12 (Gibco, Grand Island, NY, USA #12634010) containing 0.01% DNase I (Sigma-Aldrich, St. Louis, MO, USA #DN25). Digested lungs were passed through 100 μm, 40 μm, and 10 μm strainers, and single cells were collected. Cells were labeled with biotin-conjugated CD16/32 (BD Biosciences #553143), CD45 (BD Biosciences #553078), and TER110 (BD Biosciences #553672) antibodies. Streptavidin-coated magnetic beads (Invitrogen, Carlsbad, CA, USA #65601) were employed to negatively select AT2 cells. To enhance the purity and viability of AT2 cells, the cells were labeled with AF488-conjugated CD326/EpCAM antibody (Biolegend, San Diego, CA, USA #118210) and 7-AAD (Invitrogen, Carlsbad, CA, USA #A1310) viability dye. Subsequently, FACS was employed to sort EpCAM^+^ and 7-AAD^−^ AT2 cells, ensuring a collection with over 98% viability and purity. FlowJo v10.9 software was used to analyze FACS data. Unstained and single-color-stained samples were used as controls for gating.

### 4.3. Feeder-Free Culture of Alveolar Organoids

Feeder-free alveolar organoids were cultured as published previously [40]. EpCAM^+^ AT2 cells were combined with growth factor-reduced Matrigel (Corning Inc., Corning, NY, USA #354230), diluted 1:1 with organoid growth medium (AMM) [40]. Then, 60 μL of the cell suspension (30–100 cells/μL) was added to the apical chamber of transwell inserts (Corning Inc., Corning, NY, USA #3470), or 150 μL suspension was added into a well of a 6-well plate, to form 10 droplets. After a 30 min incubation at 37 °C for the matrix to solidify, 500 µL or 1500 μL of AMM with 10ng/mL IL-1β (Biolegend, San Diego, CA, USA #575102) and 10µM Y-27632 (Selleckchem, PA, USA #S1049) was introduced into the bottom well of the transwell, or into the well of the 6-well plate. Following four days of incubation, the medium was switched to AMM without IL-1β and Y-27632 and subsequently changed every 3 days. For AT1 differentiation, AMM was substituted with organoid differentiation medium (ADM) on the tenth day [40]. After a 10-day proliferation period and a 7-day differentiation phase, colonies were observed, and brightfield images were captured using an Evos XL core microscope (Life Technologies, Carlsbad, CA, USA) with a 2× microscope objective. The colony number and colony-forming efficiency of organoids were analyzed using ImageJ software [41]. Organoids were counted manually using the multi-point plugin in ImageJ. The CFE was calculated by dividing the number of colonies formed in each well by the number of cells initially seeded in the well (2000) and multiplying by 100. Five wells per group were analyzed for their organoid counts. Each dot in the scatter dot plot represents one well.

### 4.4. Western Blotting

Western blotting was performed using standard methods. Protein lysates were prepared from lung tissue, and organoids or AT2 cells in ice cold RIPA buffer and protein concentrations were determined using a BCA protein assay (Pierce, ThermoFisher Scientific, Waltham, MA, USA #23225). The protein samples were then separated using SDS-PAGE gel. Proteins were transferred to PVDF membranes and detected with anti-SPB rabbit polyclonal antibody (ThermoFisher Scientific, Waltham, MA, USA #PA5-42000, 1:1000), anti-PDPN Syrian hamster mouse antibody (Invitrogen, Carlsbad, CA, USA #MA516113, 1:1000), anti-proSPC rabbit antibody (Millipore, Bedford, MA, USA #AB3786, 1:500) and anti-β-actin mouse monoclonal antibody (Santa Cruz Biotechnology, Dallas, TX, USA #sc-47778, 1:1000). HRP-conjugated goat anti-mouse IgG (Jackson ImmunoResearch, West Grove, PA, USA #115-035-147, 1:10,000), mouse anti-rabbit IgG (Jackson ImmunoResearch, West Grove, PA, USA #211-032-171, 1:10,000), and goat anti-Syrian hamster IgG (Jackson ImmunoResearch, West Grove, PA, USA #107-035-142, 1:10,000) were used as the secondary antibodies. Blots were visualized with chemiluminescence (Millipore, Bedford, MA, USA #WBKLS0500) using a Bio-Rad Chemidoc imaging system (Bio-Rad, CA, USA). Images were analyzed using the Gel plugin under the Analyze tab in ImageJ software. Briefly, protein bands in the images were selected, and lanes were plotted. The Straight Line tool was used to connect the bottoms of the plotted lanes. Finally, the Wand tool was used to quantify the area of the plotted lanes. The band density values of the marker protein were normalized to β-actin. Results were plotted using GraphPad Prism 10 software.

### 4.5. Immunofluorescence Staining

Immunofluorescence staining and imaging were performed using standard methods. Briefly, organoids were fixed in 4% paraformaldehyde for 1 h, and then blocked and permeabilized with a solution containing 3% BSA (Sigma-Aldrich, St. Louis, MO, USA #A2153), 0.3% Triton x100 (Sigma-Aldrich, St. Louis, MO, USA #T8787), and 5% goat serum (Gibco, Grand Island, NY, USA #PCN5000). Then, organoids were incubated with primary antibodies specific to AT2 and AT1 markers, including anti-SPB rabbit polyclonal antibody (Invitrogen, Carlsbad, CA, USA #PA5-42000, 1:200) and anti-PDPN Syrian hamster mouse antibody (Invitrogen, Carlsbad, CA, USA #MA516113, 1:500). Fluorochrome-conjugated secondary antibodies, AF 488 goat anti-rabbit (Jackson ImmunoResearch, PA, USA #111-545-045, 1:500) and AF647 goat anti-Syrian hamster (Invitrogen, Carlsbad, CA, USA #A-21451, 1:1000) were used to detect primary antibodies. Finally, DAPI (1:1000) was added to detect nuclei. The organoids were mounted on slides with a spacer using a self-setting aqueous mounting medium (Electron Microscopy Sciences, PA, USA Cat #17985-10). Images were captured using a TCS5 Leica multiphoton confocal microscope (Leica Camera AG, Wetzlar, Germany). Subsequently, all images were processed and analyzed using ImageJ software.

### 4.6. Flow Cytometry

Organoids were dissociated enzymatically with TrypLETM Select (Gibco, Grand Island, NY, USA #12563-029) and passed through a 40 µm cell strainer to prepare a single-cell suspension. Cells were incubated with anti-SPB rabbit polyclonal antibody (Thermo Fisher Scientific, Waltham, MA, USA #PA5-42000, 1 µg/million cells) and AF647-conjugated anti-RAGE antibody (R & D systems, Minneapolis, MN, USA FAB11795R, 1 µg/million cells). AF488-conjugated goat anti-rabbit secondary antibody (Jackson ImmunoResearch, West Grove, PA, USA #111-545-045) was used to detect the SPB signal. Antibody-labeled cells were analyzed on a BD Fortessa flow cytometer (BD Biosciences, San Jose, CA, USA). Unstained and single-color-stained samples were used as controls for gating. Data were analyzed using FlowJo v10.9 software.

### 4.7. EdU Labeling of Proliferating AT2 Cells

To identify actively proliferating AT2 cells in organoids, we performed an EdU incorporation assay using a Click-iT 5–ethynyl–2–deoxyuridine (EdU) kit (Invitrogen, Carlsbad, CA, USA C10499). EdU (10 mM) was added to the organoid culture for 3 h, and then organoids were fixed with 4% paraformaldehyde. Then, organoids were permeabilized and processed for subsequent staining, following the kit’s instructions. Confocal microscope images were captured and analyzed to determine the percentages of EdU-positive cells. The z-sections of an entire organoid were stacked to count the total (Hoechst-stained nucleus) and EdU^+^ cells (AF488 signal) using the multi-point plugin for ImageJ. The percentage (%) of EDU+ cells was calculated by dividing the number of EDU+ cells by the total number of cells in that organoid and multiplying by 100.

### 4.8. Quantitative Reverse Transcriptase PCR

We analyzed the expression levels of CD74, HSP90, and RPS3A1 mRNA in 3D organoids using qRT-PCR. Total RNA was isolated using RNeasy Micro Kit (Qiagen, Qiagen, Hilden, Germany #74004) from organoid cultures, and 1 μg RNA was used to prepare cDNA using iScript™ Reverse Transcription Supermix (Bio-Rad, CA, USA #1708840). SYBR chemistry (Bio-Rad, CA, USA #1725270) was used to detect gene amplification. GAPDH expression was used as an internal control. Primer sequences are given in Table 1, below. Fold changes were analyzed using the delta-delta CT method.

### 4.9. Statistics Analysis

All results are reported as mean ± SD. Comparisons between WT and *Sftpc*^L184Q^ mice were made using non-parametric 2-tailed Mann-Whitney U tests, with *p* < 0.05 considered significant. Statistical tests were conducted using GraphPad Prism10 software.

## 5. Conclusions

In conclusion, *Sftpc*^L184Q^ mutation appears to halt AT2 cells in a transitional state, affecting both AT2 proliferation and AT1 transdifferentiation. This regulation is independent of the CD74, HSP90, and RPS3A1 pathways. Immunofluorescent tracking in vivo is necessary to assess *Sftpc*^L184Q^ AT2 lineage directly. Further research into the molecular mechanisms governed by *Sftpc*^L184Q^ in regulating AT2 lineages in lung repair is needed.

## Figures and Tables

**Figure 1 ijms-25-08723-f001:**
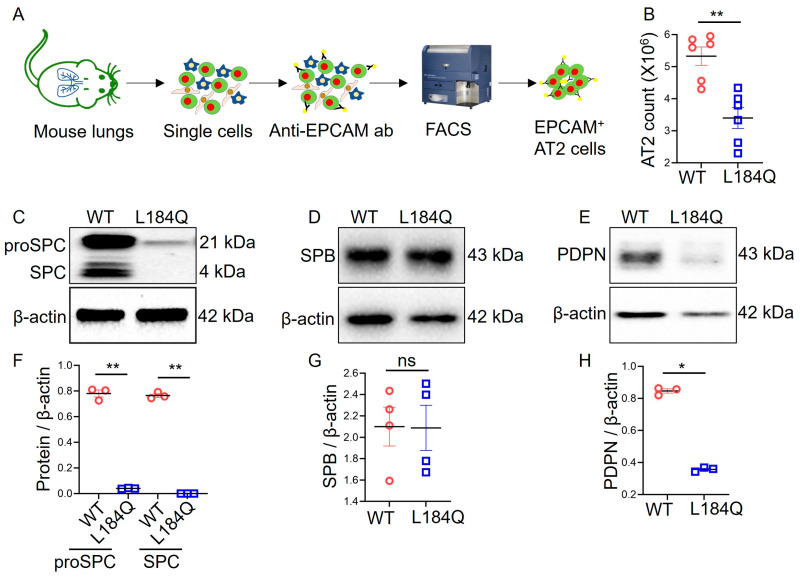
*Sftpc*^L184Q^ mutation reduced AT2 cells and cell marker proteins in vivo. (**A**) The procedure of EPCAM^+^ AT2 cell isolation using FACS; ab: antibody. (**B**) Scatter dot plot for the total AT2 population in WT (wild type, red circle) and *Sftpc*^L184Q^ (blue square) mice. Non-parametric 2-tailed Mann-Whitney U test, ** *p* < 0.01; n = 6. (**C**–**E**) Representative blots of proSPC and SPC, (**C**), SPB (**D**), and PDPN (**E**). Protein samples were prepared from whole lung lysate. (**F**–**H**) Scatter dot plot for quantification of proSPC and SPC (**F**), SPB (**G**), and PDPN (**H**). Non-parametric 2-tailed Mann-Whitney U test, * *p* < 0.05, ** *p* < 0.01; ns: not significant; n = 3–4. Data are presented as mean ± SD.

**Figure 2 ijms-25-08723-f002:**
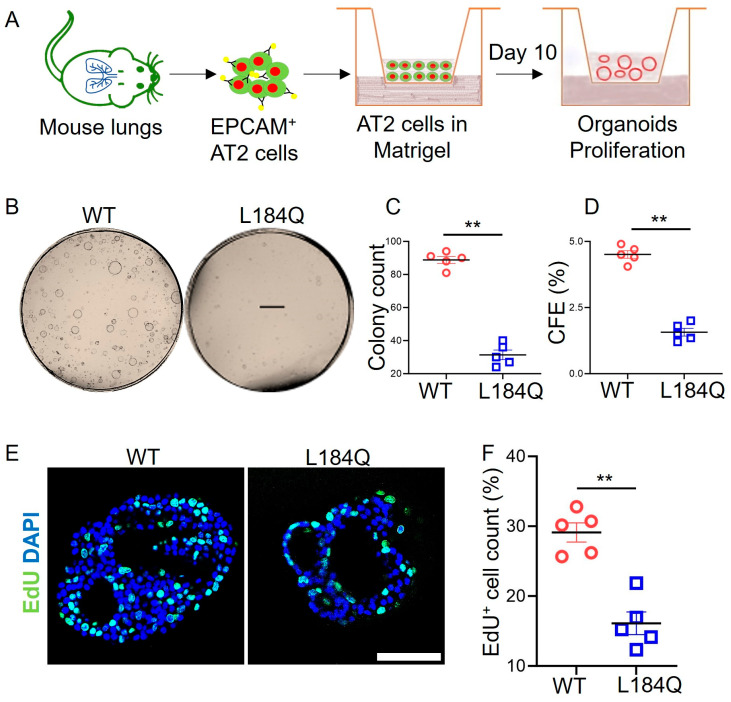
Downregulation of AT2 lineages in *Sftpc*^L184Q^ organoids. (**A**) Procedure for growing feeder-free AT2 organoid under proliferation and differentiation conditions. (**B**) Representative DIC images of proliferative organoids on day 10. Scale bar: 1 mm. (**C**) Scatter dot plot for colony count on day 10; 2-tailed Student’s *t*-test; n = 5. (**D**) Scatter dot plot for colony forming efficiency on day 10; 2-tailed Student’s *t*-test; n = 5. (**E**) Representative confocal images for EdU incorporation in proliferating AT2 cells. Scale bar: 75 µm. (**F**) Scatter dot plot comparing the numbers of EdU^+^ cells in organoids. Non-parametric 2-tailed Mann-Whitney U test, ** *p* < 0.01; n = 5. Images were analyzed using ImageJ software version 1.54j. Data are presented as mean ± SD.

**Figure 3 ijms-25-08723-f003:**
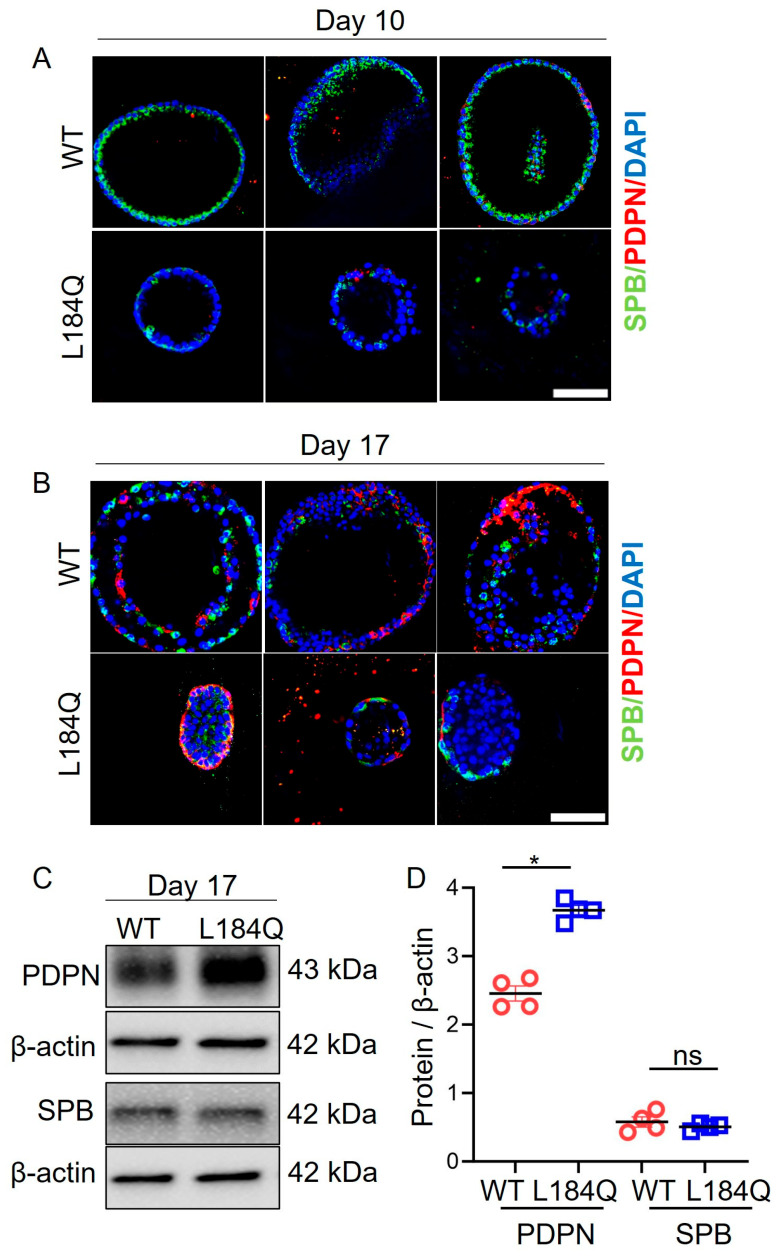
*Sftpc*^L184Q^ disrupted organoid size and structure. (**A**) Representative confocal images of 3D organoid cultures in proliferative mode on day 10. (**B**) Representative confocal images of 3D organoid cultures in differentiative mode on day 17. Scale bar: 75 µm (**C**) Western blots for AT2 (SPB) and AT1 (PDPN) cell-specific markers. (**D**) Scatter dot plot for SPB and PDPN protein levels. Non-parametric 2-tailed Mann-Whitney U test, * *p* < 0.05, ns: not significant; n = 3. Data are presented as mean ± SD.

**Figure 4 ijms-25-08723-f004:**
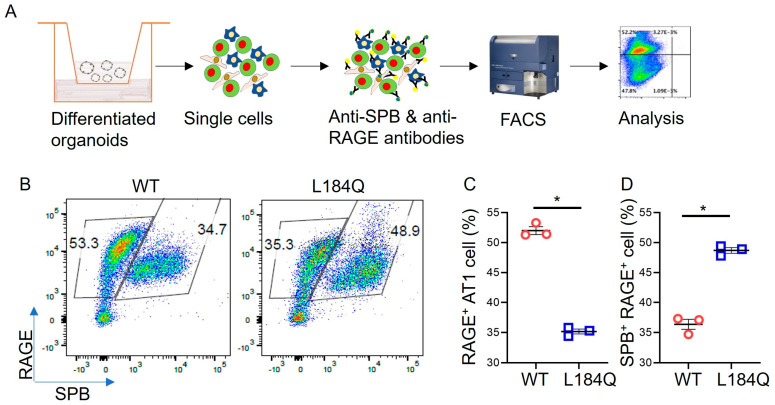
Downregulation of AT2 lineages in *Sftpc*^L184Q^ organoids. (**A**) Procedure for analyzing AT2, AT1, and transitional AT2 cell in organoids using flow cytometry. (**B**) Pseudocolour scatter dot plot for AT2, AT1, and transitional AT2 cell count in day 17 organoids. (**C**) Scatter dot plot for RAGE^+^ AT1 count. Non-parametric 2-tailed Mann-Whitney U test, * *p* < 0.05; n = 3. (**D**) Scatter dot plot for SPB^+^ RAGE^+^ transitional AT2 cell count. Non-parametric 2-tailed Mann-Whitney U test, * *p* < 0.05; n = 3. Data are presented as mean ± SD.

**Figure 5 ijms-25-08723-f005:**
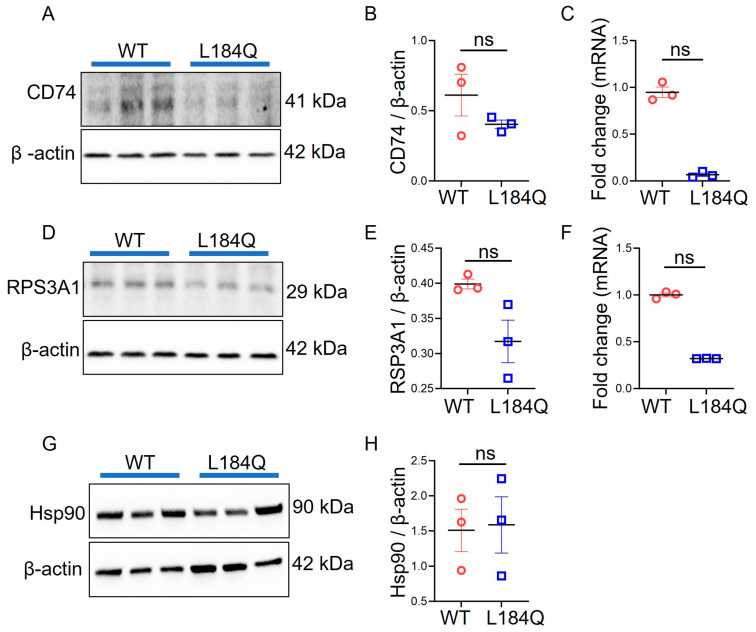
Expression levels of CD74, HSP90, and RPS3A1 proteins in organoids. (**A**) Western blot image for CD74 protein. (**B**) Scatter dot plot for levels of CD74 protein. (**C**) Scatter dot plot for CD74 mRNA expression. Non-parametric 2-tailed Mann-Whitney U test; ns: not significant; n = 3. (**D**) Western blot image for RPS3A1 protein. (**E**) Scatter dot plot for the level of RPS3A1 protein. (**F**) Scatter dot plot for Rps3a1 mRNA expression. Non-parametric 2-tailed Mann-Whitney U test; ns: not significant; n = 3. (**G**) Western blot image for HSP90 protein. (**H**) Scatter dot plot for the level of HSP90 protein. We did not run qRT-PCR for the HSP90 mRNA expression level. Non-parametric 2-tailed Mann-Whitney U test; ns: not significant; n = 3. Data are presented as mean ± SD.

**Table 1 ijms-25-08723-t001:** Primer sequences.

	Forward	Reverse
CD74	GCTGGATGAAGCAGTGGCTCTT	GATGTGGCTGACTTCTTCCTGG
RPS3A1	GGTACGATGTGAAAGCTCCAGC	GGCTCACTTCAAACACACGACC
GAPDH	CATCACTGCCACCCAGAAGACTG	ATGCCAGTGAGCTTCCCGTTCAG

## Data Availability

All data generated or analyzed during this study are included in this manuscript and its Appendix A files.

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
