# Peer review of "Humanized L184Q Mutated Surfactant Protein C Gene Alters Alveolar Type 2 Epithelial Cell Fate"

_ijms, 2024, doi:10.3390/ijms25168723_

Round 1

Reviewer 1 Report

Comments and Suggestions for Authors

Overall, this is a sound study demonstrating how SFTPC mutation affects cell fate of AT2 cells. This study could have greatly benefitted from additional data showing histological lung sections corroborating observations in the organoid system. A few comments:

1. How do authors explain a reduction in SPC due to the mutation but not SPB? Assuming that mutation in SPC leads to loss if AT2 cells, should it not decrease the overall expression of SPB as well? Visually, Fig 4A seems to show a reduction in SPB based on immunofluorescence staining. 

2. Scale bars needed in Fig 4A

3. Is there a reason authors chose different markers for AT1 for FACS (RAGE) and immunostaining (PDPN)? And have the authors longitudinally tracked the percentage of dual positive cells through the course of organoid formation?

Comments on the Quality of English Language

Needs minor improvements:

Examples:

Line 97 - 'significantly' instead of significant

Line 113 - 'regulates' instead of regulate

Line 89 - 'impairs' and 'induces' instead of impair and induce

Author Response

Dear Reviewer,

We thank you for the thoughtful comments on our manuscript (IJMS-3080215), entitled “Humanized L184Q mutated surfactant protein C gene alters alveolar type 2 epithelial cell fate”. We believe that we have addressed all concerns raised by you. Please see our point-by-point responses to the critiques in the attachment.

Thank you

Krishan

Reviewer 2 Report

Comments and Suggestions for Authors

In this study, Jain et al. invested the effect of muted Sftpc on alveolar type 2 epithelial cell using both Sftpc mutant mice as well as 3D feeder-free AT2 organoids as an in vitro model. They found that the lack of Sftpc reduced total AT2 cells in vivo and significantly attenuated colony-forming efficiency and the transition to AT1 cells using organoids.

Specific comments:

Line 101. SPB and PDPN, seen for the first time, need to spell out.

Figure 1. Can the authors please explain how they ensure that the EPCAM+ cells selected by FACS are exclusively AT2 cells, excluding other types of epithelial cells in the lungs (Fig1A and 1B)? The detection of PDPN in the sorted cell lysates (Fig 1E) suggests that the sorted cells include AT1 cells.

Figure 2. Please include a scale bar for Fig 2B. How was the CFE calculated?

Figure 3. Can the author please show the scale for X-axis for 3B? Additionally, could the authors please explain the gating strategy used to separate the right cluster into two groups? There appears to be no shift in the vertical axis for the gated regions (ie RAGE levels are the same), so the only difference seems to be in SPB expression. However, the authors have classified them into SPB+ and SPB+RAGE+ groups. Based on the data shown in 3B, there should only be two groups: RAGE+ and SPB+RAGE+.

Line 144. Please clarify “ed AT2”.

Figure 4. The authors might consider presenting figure 4 first and then figure 3. Figure 4 follows from figure 2 continuing with organoid proliferation data and moving onto differentiation. Figure 3 only has data from differentiated organoids.

Figure 5. More n needed to be conclusive, the statistics might change with more n, especially for CD74 (Fig 5B) and RSP3A1 (Fig 5E).

Line 229: Can’t find supplementary Figure E2.

Can the authors please include more details on how the images were analysis by ImageJ? Eg. Figure 2b, 2E, how many images or areas size was taken? 2E, how many cells were counted?

Comments on the Quality of English Language

Minor typos and grammatical errors. Please proofread. 

Author Response

(The authors gave the same response as above.)
